# AR-V7 in Metastatic Prostate Cancer: A Strategy beyond Redemption

**DOI:** 10.3390/ijms22115515

**Published:** 2021-05-24

**Authors:** Navid Sobhani, Praveen Kumar Neeli, Alberto D’Angelo, Matteo Pittacolo, Marianna Sirico, Ilaria Camilla Galli, Giandomenico Roviello, Gabriella Nesi

**Affiliations:** 1Department of Medicine, Section of Epidemiology and Population Sciences, Baylor College of Medicine, Houston, TX 77030, USA; navid.sobhani@bcm.edu (N.S.); praveenkumar.neeli@bcm.edu (P.K.N.); matteo.pittacolo@outlook.it (M.P.); 2Department of Biology and Biochemistry, University of Bath, Bath BA2 7AY, UK; ada43@bath.ac.uk; 3Department of Surgery and Cancer, Imperial College London, London W12 0NN, UK; mmed.sir@gmail.com; 4Azienda Socio-Sanitaria Territoriale Cremona, 26100 Cremona, Italy; 5Histopathology and Molecular Diagnostics, Careggi Teaching Hospital, 50139 Florence, Italy; galliic@aou-careggi.toscana.it; 6Department of Health Sciences, University of Florence, 50139 Florence, Italy; giandomenico.roviello@unifi.it

**Keywords:** androgen receptor, androgen receptor splice variant 7, castration-resistant prostate cancer, AR-V7 antagonists, niclosamide

## Abstract

Metastatic prostate cancer is the most common cancer in males and the fifth cause of cancer mortality worldwide. Despite the major progress in this field, leading to the approval of novel anti-androgens, the prognosis is still poor. A significant number of patients acquire an androgen receptor splice variant 7 (AR-V7), which is constitutively activated and lacks the ligand-binding domain (LBD) while maintaining the nuclear localization signal and DNA-binding domain (DBD). This conformational change, even in the absence of the ligand, allows its retention within the nucleus, where it acts as a transcription factor repressing crucial tumor suppressor genes. AR-V7 is an important oncogenic driver and plays a role as an early diagnostic and prognostic marker, as well as a therapeutic target for antagonists such as niclosamide and TAS3681. Anti-AR-V7 drugs have shown promise in recent clinical investigations on this subset of patients. This mini-review focuses on the relevance of AR-V7 in the clinical manifestations of castration-resistant prostate cancer (CRPC) and summarizes redemptive therapeutic strategies.

## 1. Introduction

Prostate cancer is the second most common cancer in men and the fifth cause of mortality worldwide, accounting for 33,330 deaths in the United States in 2020 [1]. Androgen deprivation therapy (ADT) is the cornerstone of medical management in patients with metastatic prostate cancer and biochemically recurrent disease after local treatment failure [2]. Although the use of androgen receptor signaling (ARS) inhibitors and standard chemotherapy has been shown to increase overall survival (OS) [3], drug resistance and disease progression remain a challenge [4,5,6]. According to the current European Association of Urology (EAU) guidelines, metastatic castration-resistant prostate cancer (mCRPC) is defined by biochemical progression (three consecutive rises of prostate-specific antigen (PSA) 1 week apart, resulting in two 50% increases over the nadir, with PSA > 2 ng/mL) or radiological progression (appearance of two or more bone lesions on bone scan or enlargement of a soft tissue lesion using the Response Evaluation Criteria in Solid Tumors, RECIST) with serum testosterone < 50 ng/dL [7].

In order to overcome castration resistance, the United States Food and Drug Administration (FDA) has approved the use of second-generation ADT agents targeting androgen synthesis (abiraterone acetate) and androgen receptor (enzalutamide) following favorable Phase III trial results [8,9]. Cabazitaxel, a semisynthetic tubulin-binding taxane with anti-tumor activity in docetaxel-resistant cancers, improves OS versus mitoxantrone in mCRPC patients undergoing disease progression during or after docetaxel-based therapy [10]. Given these results, cabazitaxel has been approved by the FDA in a second-line setting [11].

Data from the Phase III ALSYMPCA trial show that radium-dichloride (radium-223), an alpha emitter selectively targeting bone metastases with alpha particles [12], increases OS of CRPC patients in addition to the best standard of care [13]. The development of second-generation anti-androgen drugs has significantly expanded the armamentarium in prostatic carcinoma [14]. However, mCRPC may also acquire resistance to these compounds, in particular when abiraterone and enzalutamide are administered sequentially [15], and therefore CRPC remains a lethal disease [16].

Over the past decades, preclinical and clinical research on tumor samples from CRPC patients has uncovered diverse resistance mechanisms leading to persistent androgen receptor addiction [17]. Indeed, re-activation of the ARS pathway can occur through multiple ways: gain-of-function AR mutations, especially in the ligand-binding domain (LBD) [18]; increased intratumoral androgen biosynthesis [19,20]; AR gene amplification or overexpression [21]; crosstalk with other pathways [22,23] and upregulation of constitutively active AR splice variants [24].

Several androgen receptor splice variants have been identified in PCa cell lines and xenograft tumors at the mRNA level [25,26]. In 2010, Sun et al. developed a novel human AR splice variant in which exons 5, 6, and 7 were deleted (ARv567es) and demonstrated that this variant could contribute to cancer progression in human prostate cancer xenograft models [27]. Due to the availability of a specific antibody validated for immunohistochemistry on tumor tissue samples, androgen receptor variant 7 (AR-V7) is the best-characterized variant [28]. Since its detection in circulating tumor cells (CTCs), it has been associated with resistance to enzalutamide and abiraterone in preclinical studies [29]. AR-V7 may induce a distinctive set of gene expression compared to full-length receptor (DOI: 10.1158/0008-5472.CAN-11-3892). The ARV-7 preferentially leads to the expression of cell cycle regulatory genes, while the full-length AR represses that program and favors instead genes related to metabolism, differentiation, and macromolecular synthesis. Additionally, the resulting truncated AR variant is constitutively active and can promote AR signaling without ligand interaction. The mechanisms of cross-resistance of current anti-androgen drugs could be therefore influenced by the presence of AR-V7.

This mini-review focuses on how the structure of AR-V7 impacts its own function, leading to prostate cancer cell proliferation and resistance to second-generation hormonal therapies. In the era of precision oncology, better knowledge of AR biology, including AR splice variants, will help select mCRPC patients likely to derive little clinical benefit from potent drugs such as abiraterone and enzalutamide and offer them more effective treatment leading to a paradigm shift in mCRPC management.

## 2. Biomarkers in Advanced Prostate Cancer

Various clinical markers (e.g., performance status, number of bone/visceral metastases, time to CRPC, and previous treatments) are generally used to predict prognosis in advanced prostate cancer [30,31]. In addition, PSA, lactate dehydrogenase, alkaline phosphatase, albumin, hemoglobin, neutrophil-lymphocyte ratio, and testosterone have prognostic or predictive significance [32]. Nevertheless, these indicators are rudimentary and unable to reliably guide treatment selection, minimizing toxicity and maximizing clinical benefits. Currently, baseline CTC count is highly promising as a predictor of response [33,34].

The Gleason grading system, originally proposed by Donald Gleason in 1966, has undergone several important changes, which have been incorporated into the 2016 World Health Organization (WHO) Classification of Tumors of the Urinary System and Male Genital Organs blue book [35]. Alongside, the introduction of a prognostic Grade Group system and its ensuing validation across multiple cohorts has proven to be a remarkable advance in prostate cancer grading. Grade groupings are defined as 1 (Gleason score ≤ 6), 2 (Gleason score 3 + 4 = 7), 3 (Gleason score 4 + 3 = 7), 4 (Gleason score 4 + 4 = 8/5 + 3 = 8/3 + 5 = 8), and 5 (Gleason score 9–10) [35]. A recent meta-analysis of over 20,000 men treated by radical prostatectomy showed a strong correlation between grade groups and the risk of biochemical recurrence after surgery [36]. Considering a risk of 1 for Gleason score of 6, the relative risks of progression for grade groups 2–5 were 2.6, 8.5, 16.8, and 29.3. In grade groups 1–5, the 5-year biochemical risk-free survival accounted for 97.5%, 93.1%, 78.1%, 63.6% and 48.9%, respectively. These modifications have been crucial to maintaining the grading paradigm clinically significant and a leading prognostic factor in the contemporary management of prostate cancer.

Imaging techniques are commonly employed to assess the extent of prostate cancer spread and play a pivotal role in therapeutic decision-making [30]. Over the past few years, radionuclide bone scintigraphy (BS) and computed tomography (CT) have been the preferred methods; however, new techniques can provide more reliable detection of cancer spread [37]. Following radical prostatectomy, the combination of magnetic resonance imaging (MRI) with dynamic contrast-enhanced MRI allows for higher specificity and sensitivity than MRI alone in cancer relapse diagnosis [37]. With regard to the presence and distribution of metastatic disease, whole-body MRI displays higher specificity and sensitivity than BS or CT [38]. Recently, prostate-specific membrane antigen (PSMA) positron emission tomography (PET)/CT has been shown to be more accurate compared with conventional imaging [39].

## 3. Gold Standard Treatment of Metastatic Castration-Resistant Prostate Cancer

For many years, the cornerstone treatment of advanced prostate cancer has been traditional androgen deprivation therapy (ADT), accomplished with surgical castration or chronic administration of gonadotropin-releasing hormone analogs. This regimen is highly effective in patients with biochemical recurrence following radiation or surgical procedure for localized prostate cancer [34]; however, resistance to traditional ADT can lead to disease progression. Up until 2010, docetaxel, which has been shown to convey significant survival benefits in mCRPC, was the only regimen approved by the FDA [40]. Various new therapeutic options are now available for men diagnosed with mCRPC, including next-generation taxane-based chemotherapy (cabazitaxel), immunotherapy (sipuleucel-T), AR axis-targeted therapies (AATTs, apalutamide, enzalutamide, and abiraterone acetate), and targeted alpha-particle therapy (radium-223) [11,41,42,43,44,45]. Of these, enzalutamide and abiraterone have a better safety profile and a more favorable effect on OS compared with other FDA-approved drugs [40].

No consensus has been reached on the best treatment sequencing for mCRPC patients, nor do current clinical data support unambiguous guidance concerning the best second-line treatment. Many physicians routinely prescribe taxanes in this setting, although a substantial subgroup of patients may still benefit from oral, less toxic, and better tolerated ARS inhibitors. In addition, an empiric sequencing approach could result in toxicity, higher treatment costs, and delays in patients administered with potentially more effective regimens [46,47].

## 4. Outcomes and Survival in Castration-Resistant Prostate Cancer

In recent years, AR-V7 genomic alterations have been associated with progression to mCRPC, patients with detectable AR-V7 in liquid biopsies usually manifesting more aggressive disease and shorter survival [48]. Emerging evidence suggests that AR-V7 status can act as a prognostic marker in mCRPC [49]. More aggressive features (e.g., worse performance status, higher PSA serum levels, and higher disease burden) and poorer clinical outcomes have been observed in AR-V7-positive patients [48].

Both preclinical and clinical studies have demonstrated a correlation between abiraterone or enzalutamide resistance and AR-V7 expression, as well as decreased OS in CRPC patients [50,51,52]. Indeed, when administered ARS inhibitors, AR-V7-positive patients have a shorter OS and progression-free survival (PFS) compared with AR-V7-negative patients. Scher et al. reported longer survival in AR-V7-negative patients under taxane-containing regimens [53]. These authors also showed longer OS (median 8.9 vs. 4.6 months) and PFS (median 5.3 vs. 2.3 months) in AR-V7-positive mCRPC patients receiving taxanes compared with those on abiraterone and enzalutamide [53]. Conversely, two prospective trials found that AR-V7 status has no substantial impact on OS in patients administered any chemotherapy regimen [54,55]. Similar outcomes were recorded in AR-V7-positive mCRPC patients receiving abiraterone (median OS 10.6 months, median PFS 2.3 months), enzalutamide (median OS 5.5 months, median PFS 2.1 months), and taxanes (median OS 9.2 months, median PFS 5.1 months) [51,54].

## 5. AR-V7 Structure and Function

The canonical AR gene, located on the X chromosome (Xq11-12), exceeds 90 kb and contains eight exons encoding the AR protein, a ligand-dependent nuclear transcription factor and member of the steroid hormone receptor family [56]. AR is a 110 kD protein consisting of 919 amino acids and four functional domains: (1) the N-terminal transactivation domain (NTD); (2) the DNA-binding domain (DBD); (3) the hinge region, and (4) the LBD [57,58,59]. Exon 1 of wild-type AR codes for the NTD, which contains 537 amino acids (1–537 region). NTD is constitutively active, facilitates transcriptional activation function-1 (AF-1), and plays an essential role in forming the cellular transcription complex. AF-1 comprises two transactivation units, TAU-1 (142–485 amino acids) and TAU-5 (351–528 amino acids) [60]. TAU-5 regulates the constitutive transcriptional activity and is responsible for aberrant activation of AR in CRPC cells [61,62,63]. Exons 2 and 3 encode two zinc fingers in the DBD (538–624 amino acids) that recognize specific DNA sequences and promote AR homodimerization. Exon 4 encodes the hinge region (625–669 amino acids), which separates the DBD from the LBD and contains the nuclear localization signal (NLS) required for AR nuclear translocation. Exons 5–8 encode the LBD (626–919 amino acids), which harbors the AF-2 and enables binding of androgen ligands, the primary control mechanism of the ARS axis (Figure 1) [57,61].

AR-V7 mRNA retains the first three canonical exons, followed by variant-specific cryptic exon 3 (CE3) within intron 3 [27,64,65]. A splicing event at CE3 leads to an LBD-truncated AR-V7 owing to premature translation termination after 16 variant-specific amino acids [64]. To detect the full-length AR (AR-FL) and AR-V7 mRNA, quantitative reverse-transcriptase-polymerase-chain-reaction (qRT-PCR) can be performed with primer sets spanning specific splice junctions [66]. Antonarakis et al. identified AR-V7 mRNA in CTCs from CRPC patients and demonstrated that the presence of AR-V7 in neoplastic cells is associated with both enzalutamide and abiraterone resistance [51]. This may be due to AR-V7 lacking the LBD, direct target of enzalutamide, and indirect target of abiraterone, though remaining constitutively active in a ligand-independent manner [28,67]. Compounds that degrade the AR, e.g., niclosamide and galeterone, or prevent AR-mediated transcription, e.g., bromodomain-containing protein 4 (BRD4) inhibitors, are promising therapeutic options in patients progressing after abiraterone and/or enzalutamide [68,69,70] (Figure 2). Proteolysis targeting chimeras (PROTACs), also known as protein degraders, are a new class of small molecule therapeutics [71]. They differ from traditional inhibitors or antagonists in that they use a dual-binding system within each cancer cell to remove unwanted or damaged proteins. PROTACs selectively bind to the target proteins as well as to an E3 ubiquitin ligase, which in turn triggers the ubiquitination system tagging the protein for proteosomal degradation. With no need to bind to the target protein active site, PROTAC could be beneficial as it bypasses resistance to conventional therapies for prostate cancer and it is recycled by the cell.

## 6. From Alterations to AR Axis-Targeted Therapy Resistance

Numerous molecular alterations are reported to be associated with AR Axis-Targeted Therapy (AATT) resistance [72]. Although mCRPC is more likely to occur following ADT, the majority of prostatic adenocarcinomas remain reliant on ARS. Such mechanisms comprise: (1) steroidogenesis upregulation within the prostate tumor, allowing the synthesis of endogenous androgens [20,73,74]; (2) higher AR expression in prostate tumor cells, mainly due to AR gene amplification [22]; (3) single point mutations within AR gene LBD [75,76,77]; (4) silencing through methylation of the gene encoding the androgen-inactivation enzyme HSD17B2 [78]; (5) variants of the HSD3B1 gene [79]; (6) upregulation of the glucocorticoid receptor [80], and (7) emergence of AR splice variants.

Of paramount importance is the ability of AR-V7 to repress the transcription of tumor suppressor genes [81]. Unlike AR-FL that involves both coactivators and corepressors, AR-V7 preferentially interacts with the nuclear receptor corepressor (NCoR), thus confirming that AR-FL and AR-V7 exhibit differential co-regulatory networks. Interestingly, the difference in AR-V7 and AR-FL transcriptional activity seems to be related to AR isoform-specific variations in H3K27 acetylation. AR-V7 interplays with corepressors (e.g., NCoR1 and NCoR2), which in turn drive the genomic recruitment of histone deacetylases (e.g., HDAC3), negative regulators of H3K27 acetylation. Following AR-V7 depletion, FOXA1 cistrome reprogramming may arise from AR inhibition, although not directly associated with AR-V7-dependent repression, also occurring at AR-V7-activated sites [81]. In line with this observation, AR-V7 and FOXA1 were found to be unable to interact directly [82,83]. The existence of other AR-V7 co-operating factors must be considered, and HOXB13 has recently been shown to be crucial in mediating AR-V7 function [84].

## 7. AR-V7 Expression and Predictive Potential

The expression of AR-V7 has been extensively investigated in both normal and neoplastic prostate tissue. Using a recombinant rabbit monoclonal antibody validated for immunohistochemistry, Sharp et al. demonstrated that AR-V7 is rarely observed in primary prostate tumor specimens (<1%) but is frequent in metastatic biopsy specimens after ADT (>75%), suggesting that AR-V7 increases under the selective pressure of hormonal treatment [85]. AR-V7 protein expression was predominantly seen in the nucleus of mCRPC cells and was widely heterogeneous in different metastases from the same patient. Furthermore, AR-V7 negative patients showed better PSA response rates (100% vs. 54%; *p* = 0.03) and higher OS than AR-V7 positive patients (74 vs. 24 months; hazard ratio [HR], 0.23; *p* = 0.02).

Several studies comparing outcomes of mCRPC patients administered second- and third-line therapies have identified AR-V7 in CTCs by means of qRT-PCR and immunohistochemistry. Antonarakis et al. were the first to show that the presence of AR-V7 transcripts in CTCs from mCRPC patients is correlated with second-line AATT resistance [51]. Of 31 enzalutamide-treated patients and 31 abiraterone-treated patients, 39% and 19% had detectable AR-V7 in CTCs, respectively. AR-V7-positive patients showed significantly worse outcomes than AR-V7-negative patients, with a lower PSA response and shorter PFS. In a subsequent study on 202 CRPC patients taking either enzalutamide or abiraterone, these authors confirmed that AR-V7 positivity significantly correlates with other negative prognostic factors, including Gleason score ≥ 8, metastatic disease at diagnosis, and prior treatment with second-line AATT and taxanes [86].

Using automated immunofluorescence staining, Scher et al. investigated the association of clinical outcome with intranuclear AR-V7 protein in CTCs from 161 mCRPC patients undergoing a change in their treatment [53]. AR-V7 positive CTCs were detected in 34 samples (18%), which included 3% of first-line, 18% of second-line, and 31% of third- or subsequent lines of therapy (*p* < 0.001). During AATT, OS and PFS were significantly shorter in AR-V7-positive than in AR-V7-negative patients. In AR-V7-positive patients, multivariate analysis revealed that HR for death was lower in those treated with taxanes compared with those treated with AATT (HR, 0.24; *p* = 0.035). This study concluded that: (1) AR-V7 positivity increases with the number of treatment lines; (2) AR-V7 positive CTCs can predict a lack in clinical AATT benefit and (3) AR-V7 positive CTCs in men with mCRPC can predict resistance to ARS inhibitors but not to taxanes [53].

A follow-up analysis by the same authors confirmed that nuclear-specific AR-V7 protein is a valuable biomarker and is strongly associated with survival [87]. These findings were further validated in a multicenter study performed on 142 mCRPC patients receiving either taxanes or ARS inhibitors [88]. Pre-therapy CTC nuclear expression of AR-V7 protein correlated with higher OS for taxane rather than ARS inhibitor therapy (*p* = 0.03). Nuclear localized AR-V7 assay can therefore assist treatment decision-making, ensuring maximum patient benefit [88].

## 8. Ongoing Clinical Trials

Clinical trials currently analyzing AR-V7 antagonists consist of two investigations on niclosamide and one on TAS3681. Niclosamide is being tested in combination with enzalutamide (NCT03123978) and in combination with abiraterone plus prednisone (NCT02807805).

The main aim of the NCT03123978 Phase I clinical trial is to define safety and recommend the correct dose for the Phase II clinical trial, while the secondary goal is to determine PFS and PSA response. With regard to NCT02807805 Phase II clinical trial, the primary objective is to investigate PSA response to the combination of drugs, and the secondary objectives encompass overall response rate, PFS, and toxicity. NCT02566772 Phase I clinical trial aims to set the dose-limiting toxicity (DLT) of TAS3681, a drug down-regulating the full-length and splice variants of AR.

Niclosamide is specific for AR-V7, and results from these clinical trials may disclose the validity of targeting AR-V7 alone in CRPC. Contrariwise, TAS3681 could be used to target both AR-FL and AR-V7 in the treatment of CRPC with the same formula. Once safety and tolerability of niclosamide and TAS3681 have been verified, further trials can be conducted to evaluate their efficacy. After determining safety and tolerability, randomized trials may be conducted to further evaluate the efficacy of niclosamide and TAS3681.

Recently, the PROTAC ARV-110 has also been tested in a Phase I/II open label, single-group clinical trial (NCT03888612), consisting of 50 mCRPC participants. EPI-001 was the first reported AR amino-terminal domain (NTD) inhibitor, to block the protein-protein interaction triggering transcriptional activity of AR and its splice variants [doi:10.1093/annonc/mdu038; doi:10.1016/j.ccr.2010.04.027], differently from conventional therapies targeting the ligand-binding domain. For this reason, EPI-001 was initially considered useful for mCRPC treatment. The successor to EPI-001, EPI-506, a prodrug of ralaniten (EPI-002), was investigated in a Phase I study on mCRPC, but was discontinued after showing only minor PSA decline. Though well tolerated, it proved ineffective probably on account of its high metabolization rate, indeed, 19 metabolites were detected in patient plasma [doi:10.1200/JCO.2019.37.7_suppl.257 Journal of Clinical Oncology 37, no. 7_suppl (1 March 2019) 257-257]. AR-V7 antagonists, niclosamide and TAS3681, and possibly PROTAC ARV-110, therefore remain the most promising drugs for mCRPC patients progressing under second-generation androgen-depriving therapeutics. Table 1 summarizes ongoing clinical investigations testing AR-V7 antagonists.

## 9. Conclusions

Despite recent therapeutic advances, CRPC still remains a lethal disease. AR-V7 is a constitutively active AR isoform that does not require ligand binding and is difficult to treat with commercial androgen-depriving second-generation drugs. AR-V7-positive CTCs have been associated with poor clinical outcome and AATT resistance, thus making AR-V7 a valid biomarker to estimate patient prognosis. Novel drugs such as niclosamide and TAS3681, capable of targeting this variant, have been developed. Future randomized and large clinical trials testing these molecules in patients with AR-V7 positive CRPC are warranted.

## Figures and Tables

**Figure 1 ijms-22-05515-f001:**
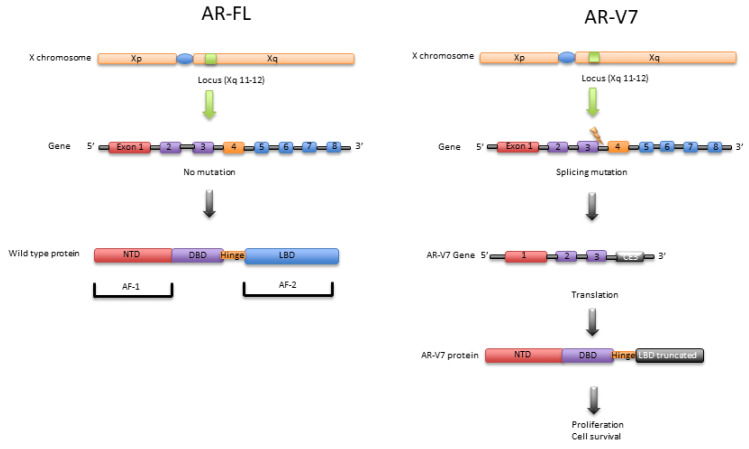
Transcript structures for the full-length androgen receptor (AR-FL) and splice variant AR-V7.

**Figure 2 ijms-22-05515-f002:**
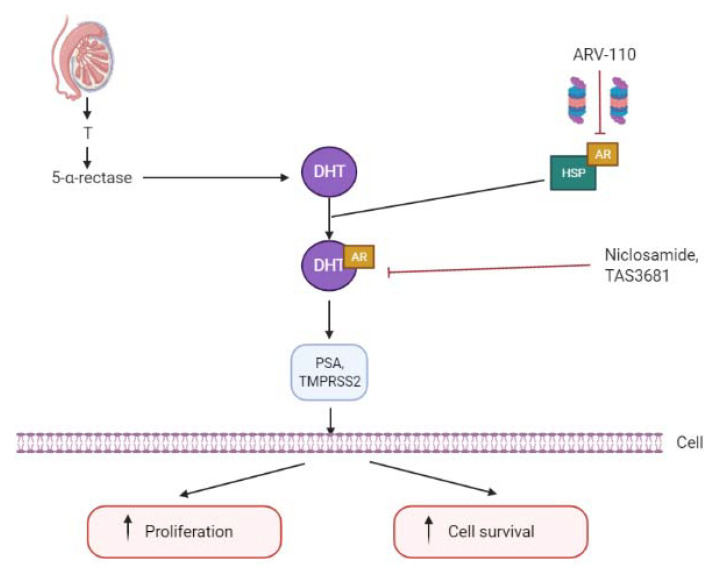
Molecular pathways of AR-V7 antagonists on full-length androgen receptor (AR-FL) and AR-V7.

**Table 1 ijms-22-05515-t001:** Ongoing clinical trials with AR-V7 antagonists in CRPC.

Clinical Trial ID	Phase	Drugs	Primary Endpoint	Status
NCT02807805	II	Niclosamide, abiraterone acetate and prednisone	PSA response	Recruiting
NCT03123978	I	Niclosamide and enzalutamide	PFS, DLT, RP2D	Recruiting
NCT02566772	I	TAS3681	DLT	Recruiting
NCT03888612	I/II	ARV-110	DLT	Recruiting

(AR-V7: androgen receptor splice variant 7; CRPC: castration-resistant prostate cancer; PSA: prostate-specific antigen; PFS: progression-free survival; DLT: dose-limiting toxicity; RP2D: recommended Phase II dose).

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
