# Peer review of "AR-V7 in Metastatic Prostate Cancer: A Strategy beyond Redemption"

_ijms, 2021, doi:10.3390/ijms22115515_

Round 1
Reviewer 1 Report
This review describes in a concise but very precise way the role of the presence of AR-V7 in the survival and response to treatment in prostate cancer. This review is extremely well written, very well documented with the citation of numerous articles and some very recent. English is perfect. The only small criticism, if one has to be made, is that Figure 2 is really very simple. It deserved to be more detailed. This review will benefit, I am sure, all clinicians and researchers interested in prostate cancer and its treatment.
Author Response
We added a PROTAC to the graph. We tried to not add much because we want to keep it nice and simple.
Reviewer 2 Report
Dear Authors,
Sobhani et al, is a well-written mini-review concerning AR-V7, a major culprit behind mCRPC and an obstacle in prostate cancer recovery. The authors do an excellent job presenting the current gold standard of PCa treatment, but also highlight that there is no consensus regarding the disease management upon CRPC emergence. PCa patients desperately need predictive biomarkers to design personalized treatment strategies, but also novel therapies that overcome acquired drug-resistance. The authors touch upon all these issues and discuss a few ongoing clinical trials with 2 promising compounds.
AR-V7 in metastatic prostate cancer is a timely topic of great interest to Cancers audience. Sobhani et al has a great potential to benefit the prostate cancer field, however there are 3 key aspects concerning AR-V7 that this manuscript is currently missing but would benefit from:
1) In lines 58-60 the authors mention ‘mCRPC may also acquire resistance to these compounds, in particular when the two agents are administered sequentially’. Perhaps the authors can provide a specific example? Also, either here in the introduction or later in section 6 (lines 205-228) the authors should discuss drug cross-resistance e.g. like the one seen between Enzalutamide and Abiraterone; and if AR-V7 could be involved/responsible.
2) In lines 196-199 and in Figure 2 the authors discuss AR degradation, but with the development of SARDs and PROTACs, this topic is hardly exhausted. In particular, it would be beneficial to discuss recent observations indicating that targeting AR-WT for degradation presents a therapeutic potential even when AR-V7 is present, suggesting that AR-V7 might require AR-WT to drive cancer progression.
3) In section 8 (lines 271-288) the authors discuss ongoing clinical trials concerning AR-V7 with a particular focus on Niclosamide and TAS3681. What about compounds targeting N-terminus of AR that are predicted to affect AR-FL and AR splice variants alike e.g. EPI-001 (and derivatives)? Is targeting AR N-terminus a viable strategy to alleviate AR-V7’s mCRPC-driving potential?
Other minor aspects of the paper that require clarification/correction:
4) Line 19: the authors indicate that many patients ‘inherit’ AR-V7. Clarification might be needed whether the authors suggest a germline AR-V7 mutation in being passed on from the mother onto her male offspring rather than being acquired with the disease development as indicated in lines 233-235.
5) In lines 186-187 the authors indicate ‘AR-V7 mRNA retains the first three canonical exons, followed by variant-specific 186 cryptic exon 3 (CE3) within intron 3’. However, in Figure 1, AR-V7 gene exons 2 and 3 are labeled in blue and appear to be longer than the purple equivalents of AR-FL indicating they are altered. For the benefit of the audience, it would be important to ensure consistency and label the exact mutation site where the alteration occurs. Likewise, AR-V7 protein, NTD and DBD appear to be much longer than in AR-FL.
6) Line 206: please consider revising ‘AATT’ to ‘AR Axis-Targeted Therapy (AATT)’ should it be the intended meaning.
7) Line 217: please consider revising ‘NCOR’ to ‘Nuclear receptor corepressor (NCoR)’ should it be the intended meaning.
Author Response
Sobhani et al, is a well-written mini-review concerning AR-V7, a major culprit behind mCRPC and an obstacle in prostate cancer recovery. The authors do an excellent job presenting the current gold standard of PCa treatment, but also highlight that there is no consensus regarding the disease management upon CRPC emergence. PCa patients desperately need predictive biomarkers to design personalized treatment strategies, but also novel therapies that overcome acquired drug-resistance. The authors touch upon all these issues and discuss a few ongoing clinical trials with 2 promising compounds.
AR-V7 in metastatic prostate cancer is a timely topic of great interest to Cancers audience. Sobhani et al has a great potential to benefit the prostate cancer field, however there are 3 key aspects concerning AR-V7 that this manuscript is currently missing but would benefit from:
- In lines 58-60 the authors mention ‘mCRPC may also acquire resistance to these compounds, in particular when the two agents are administered sequentially’. Perhaps the authors can provide a specific example?
Thank you. We have added the example of abiraterone and enzalutamide.
Also, either here in the introduction or later in section 6 (lines 205-228) the authors should discuss drug cross-resistance e.g. like the one seen between Enzalutamide and Abiraterone; and if AR-V7 could be involved/responsible.
We explained it in the introduction.
- In lines 196-199 and in Figure 2 the authors discuss AR degradation, but with the development of SARDs and PROTACs, this topic is hardly exhausted. In particular, it would be beneficial to discuss recent observations indicating that targeting AR-WT for degradation presents a therapeutic potential even when AR-V7 is present, suggesting that AR-V7 might require AR-WT to drive cancer progression.
We have added a discussion on the therapeutic potential of PROTAC and added a clinical trial too.
- In section 8 (lines 271-288) the authors discuss ongoing clinical trials concerning AR-V7 with a particular focus on Niclosamide and TAS3681. What about compounds targeting N-terminus of AR that are predicted to affect AR-FL and AR splice variants alike e.g. EPI-001 (and derivatives)? Is targeting AR N-terminus a viable strategy to alleviate AR-V7’s mCRPC-driving potential?
We added them in the clinical trials section. Although the type of therapy is interesting, because blocks protein-protein interaction, it has been discontinued in phase I since it was highly metabolized. For now the most promising drugs may be other ones, such as Niclosamide and TAS3681.
Other minor aspects of the paper that require clarification/correction:
- Line 19: the authors indicate that many patients ‘inherit’ AR-V7. Clarification might be needed whether the authors suggest a germline AR-V7 mutation in being passed on from the mother onto her male offspring rather than being acquired with the disease development as indicated in lines 233-235.
We fixed it.
5) In lines 186-187 the authors indicate ‘AR-V7 mRNA retains the first three canonical exons, followed by variant-specific 186 cryptic exon 3 (CE3) within intron 3’. However, in Figure 1, AR-V7 gene exons 2 and 3 are labeled in blue and appear to be longer than the purple equivalents of AR-FL indicating they are altered. For the benefit of the audience, it would be important to ensure consistency and label the exact mutation site where the alteration occurs. Likewise, AR-V7 protein, NTD and DBD appear to be much longer than in AR-FL.
We fixed it.
6) Line 206: please consider revising ‘AATT’ to ‘AR Axis-Targeted Therapy (AATT)’ should it be the intended meaning.
Thank you. We added the definition.
7) Line 217: please consider revising ‘NCOR’ to ‘Nuclear receptor corepressor (NCoR)’ should it be the intended meaning.
Thank you. We added the definition.